# Differentially Expressed Genes Associated with the Development of Cervical Cancer

**DOI:** 10.3390/ijms27010258

**Published:** 2025-12-26

**Authors:** Diego Armando Alvarado-Camacho, Ricardo Castillo-Velázquez, Angelica Judith Granados-López, Hiram Hernández-López, Yamilé López-Hernández, Rosalinda Gutiérrez-Hernández, José Antonio Varela-Silva, Claudia Araceli Reyes-Estrada, Cesar Rogelio Solorio-Alvarado, Sergio Hugo Sánchez-Rodríguez, David Alejandro García-López, Jesús Adrián López

**Affiliations:** 1Laboratorio de microRNAs y Cáncer, Unidad Académica de Ciencias Biológicas, Universidad Autónoma de Zacatecas “Francisco García Salinas”, Av. Preparatoria S/N, Zacatecas 98066, Mexico; diego_arm1320@hotmail.com (D.A.A.-C.); agranados@uaz.edu.mx (A.J.G.-L.); jose.varela@uaz.edu.mx (J.A.V.-S.); 2Unidad de Investigación Biomédica de Zacatecas, Instituto Mexicano del Seguro Social, Zacatecas 98000, Mexico; mcytq.ricardo@gmail.com; 3Laboratorio de Síntesis Química, Unidad Académica de Ciencias Químicas, Campus Siglo XXI, Universidad Autónoma de Zacatecas “Francisco García Salinas”, Kilometro 6, Ejido la Escondida, Zacatecas 98160, Mexico; hiram.hernandez.lopez@uaz.edu.mx; 4Laboratorio de Proteómica y Metabolómica, Unidad Académica de Ciencias Biológicas, Universidad Autónoma de Zacatecas “Francisco García Salinas”, Av. Preparatoria S/N, Zacatecas 98066, Mexico; ylopezher@conacyt.mx; 5Laboratorio de Etnofarmacología y Nutrición, Unidad Académica de Nutrición, Campus Siglo XXI, Universidad Autónoma de Zacatecas “Francisco García Salinas”, Kilometro 6, Ejido la Escondida, Zacatecas 98160, Mexico; rosalinda@uaz.edu.mx; 6Laboratorio de Inmunohistoquímica y Estrés Oxidativo, Unidad Académica de Medicina Humana, Campus Siglo XXI, Universidad Autónoma de Zacatecas “Francisco García Salinas”, Kilometro 6, Ejido la Escondida, Zacatecas 98160, Mexico; c_reyes13@yahoo.com.mx; 7División de Ciencias Naturales y Exactas, Departamento de Química, Campus Guanajuato, Universidad de Guanajuato, Noria Alta S/N, Guanajuato 36050, Mexico; csolorio@ugto.mx; 8Laboratorio de Biología Celular y Neurobiología, Unidad Académica de Ciencias Biológicas, Universidad Autónoma de Zacatecas “Francisco García Salinas”, Av. Preparatoria S/N, Zacatecas 98066, Mexico; smdck@hotmail.com (S.H.S.-R.); davidrockerdagal@uaz.edu.mx (D.A.G.-L.)

**Keywords:** cervical cancer, microarray, gene expression, mRNAs, microRNAs

## Abstract

Cervical cancer remains a significant cause of cancer-related mortality among women, particularly in low- and middle-income countries. High-throughput technologies, such as microarrays, have facilitated the comprehensive analysis of gene expression profiles in cervical cancer, enabling the identification of key differentially expressed genes (DEGs) involved in its pathogenesis. The publicly available microarray datasets, including GSE39001, GSE9750, GSE7803, GSE6791, GSE63514, and GSE52903 in combination with bioinformatics database predictions, were used to identify differential expression genes, potential biomarkers, and therapeutic targets for cervical cancer; additionally, we undertook bioinformatic analysis to determine gene ontology and possible miRNA targets related to our DEGs. Our analysis revealed several DEGs significantly associated with cervical cancer progression, such as cell death, regulation of DNA replication, protein binding processes, and transcription factors. The most relevant transcription factors (TFs) identified were SP1, ELF3, E2F1, TP53, RELA, HDAC, and FOXM1. Importantly, the DEGs with more important changes were 11 coding genes that were upregulated (*KIF4A, MCM5, RFC4, PLOD2, MMP12, PRC1, TOP2A, MCM2, RAD51AP1, KIF20A, AIM2*) and 14 that were downregulated (*CXCL14, KRT1, KRT13, MAL, SPINK5, EMP1, CRISP3, ALOX12, CRNN, SPRR3, PPP1R3C, IVL, CFD, CRCT1*), which were associated with cervical cancer. Interestingly, hub proteins KIF4A, NUSAP1, BUB1B, CEP55, DLGAP5, NCAPG, CDK1, MELK, KIF11, and KIF20A were found to be potentially regulated by several miRNAs, including miR-107, miR-124-3p, miR-147a, miR-16-5p, miR-34a-5p, miR-34c-5p, miR-126-3p, miR-10b-5p, miR-23b-3p, miR-200b-3p, miR-138-5p, miR-203a-3p, miR-214-3p, and let-7b-5p. The relationship between these genes highlights their potential as candidate biomarkers for further research in treatment, diagnosis, and prognosis.

## 1. Introduction

Cervical cancer ranks as the fourth most common cancer among women globally, with approximately 661,021 new cases and 348,189 deaths reported in 2022, according to the Global Cancer Observatory (GLOBOCAN) [1]. Despite improvements in screening and vaccination programs, cervical cancer remains a significant public health challenge, particularly in regions with limited healthcare access. Persistent infection with high-risk human papillomavirus (HPV) types is the primary driver of cervical cancer, leading to genetic and epigenetic alterations that disrupt normal cellular function and promote malignant transformation [2]. HPV propagation is dependent on the cellular differentiation state and the abundance of important molecules, including transcriptional factors, polymerases, splicing factors, RNA processing machinery, and regulatory elements of RNAs [3]. HPV infection results in the expression of viral proteins that lead to the acquisition of immortal cell features, such as sustained proliferation and differentiation. HPVs, such as HPV16 and HPV18, are associated with oncogenesis and are therefore considered to be “high risk” (HR) viruses. The HR viral E7 oncoprotein interacts with retinoblastoma (Rb) protein family members, releasing the transcriptional factor E2F from Rb regulation, permitting the transition from G1 to S phase [4]. The binding of E2F to the target promoter activates transcription of several genes involved in proliferation [5]. In addition, the HR-HPV E6 protein interacts with p53 and induces its degradation via ubiquitination, resulting in a p53-null phenotype that abrogates apoptosis and cell cycle checkpoints [3,6]. P53 is altered by several stimuli, such as UV exposure, oncogene activation, and replications stress, among several other factors [7,8]. Additionally, it has been reported that HPV integration can induce gene expression misregulation. Besides the misregulation of coding genes by HR-HPV oncoprotein expression, HPV integration is associated with non-coding genes like microRNAs’ (miRNAs) dysregulation [9]. Therefore, cervical cancer is a complex genetic pathology that involves coding gene and non-coding gene abnormalities [10] that are also involved in tumor cell microenvironment modulation [11]. Normal cellular proliferation is regulated by proto-oncogenes and tumor gene suppressors. Mutations that increase proto-oncogene activity convert these genes into oncogenes, leading to tumor cell proliferation. Normally, these genes encode growth factors, growth factor receptors, transduction signaling proteins, and DNA-binding proteins, among others [12]. On the other hand, mutations that inactivate tumor gene suppressors liberate the genetic inhibition and thereby potentiate tumor cell proliferation. Cellular proliferation is not an autonomic event; it is regulated by intercellular communication, ensuring normal tissue integrity. Examples of intercellular signals include contact inhibition and anchorage-dependent growth, which are both hallmarks of normal cells [13]. Additionally, cell-to-cell microRNA exchange is an important means of cell communication and processes regulation [14,15,16]. miRNAs regulating oncogenes are known as anti-oncomiRs, and miRNAs inhibiting tumor gene suppressors are known as oncomiRs [10,11,12,13,14,15,16,17]. One of the most important features of miRNAs is their ability to regulate multiple mRNA targets [18], allowing for the analysis of numerous coding genes with a single miRNA [19]. MiRNAs are non-coding regulatory RNAs 19–25 nucleotides (nt) in size that are produced by RNA polymerase II (pol II) and III (pol III), derived from transcripts of coding or non-coding genes. Many miRNAs are tissue-specific or differentiation-specific, and their temporal and spatial expression patterns modulate gene expression at the post-transcriptional level by base pairing with the complementary nucleotide sequences of target mRNAs [20,21]. The sequence of partial and total complementarity binding of miRNAs to target mRNAs inhibits protein translation or degrades target mRNAs, respectively [18].

The bioinformatics prediction has indicated that each miRNA targets more than 100 RNA transcripts, and up to one-third of the total number of human mRNAs is regulated by these non-coding genes [20,21,22]. Therefore, miRNAs exert deep effects on gene expression in almost every biological process, such as proliferation, anchorage-independent growth, apoptosis, migration, and invasion [15]. Indeed, restoration of miRNA expression or miRNA inhibition alters cellular processes [23]. Therefore, miRNAs are a powerful tool for gene therapy, prognosis, and diagnosis of malignant diseases. miRNAs expression that is affected by HPV infection specifically occurs as a cause of cervical cancer [24], while some others are altered independently of HPV infection, probably as consequence of the progression of the disease. Distinguishing between the involvement of miRNAs as a consequence and/or a cause of cancer has not been solved until now; however, it is a fact that they orchestrate gene profile changes along carcinogenesis [25,26]. Genes regulated by miRNAs are of great value for cancer treatment, diagnosis, and prognosis, prompting substantial search on this topic [17,27,28]. Microarray technology has revolutionized the field of genomics, allowing researchers to simultaneously analyze the expression of thousands of genes across different conditions. This approach has been valuable in cancer research, where identifying differentially expressed genes (DEGs) between normal and tumor tissues can reveal insights into the molecular mechanisms driving cancer progression in cervical cancer [29,30,31].

In this study, we utilized several publicly available microarray datasets to perform a comprehensive analysis of DEGs in cervical cancer. Our goal was to find highly expressed genes in cervical cancer compared to healthy controls, and to determine whether these genes are regulated by specific miRNAs through a bioinformatics tool. The datasets used include GSE39001, GSE9750, GSE7803, GSE6791, GSE63514, and GSE52903. These datasets collectively encompass a wide range of cervical cancer samples and controls, allowing for a robust analysis of gene expression patterns. GSE39001, GSE7803, GSE6791, GSE63514, and GSE52903 further expand the dataset, providing additional data from different stages and types of cervical cancer, as well as controls. GSE arrays provided data from 197 cervical carcinomas and 95 control cervical tissues. By integrating data from these diverse studies, we aimed to improve the robustness of our findings and uncover novel insights into the molecular drivers of cervical cancer. Identifying key DEGs may facilitate the development of targeted therapies, ultimately improving patients’ outcomes.

## 2. Results

### 2.1. Microarray Data Information and DEGs Analysis in Cervical Cancer

According to our workflow diagram strategy (Figure 1), six GEO microarray datasets (GSE39001, GSE9750, GSE7803, GSE6791, GSE63514, and GSE52903) were selected through a systematic screening process to ensure analytical comparability, biological relevance, and data quality. The inclusion criteria required the following: (I) availability of raw or normalized expression matrices; (II) clearly annotated clinical information, including the presence of cervical cancer samples and non-cancer controls; and (III) use of well-standardized microarray platforms with reliable probe annotation. Several additional GEO series were initially examined but then excluded due to lack of matched controls, small sample sizes, incomplete metadata, or platforms with limited probe coverage, all of which hindered cross-study comparability.

The final selected datasets included both the Affymetrix and Agilent arrays: two platforms known for producing reproducible gene-expression profiles across independent studies. Since cross-platform merging can introduce artificial batch effects, each dataset was analyzed separately using GEO2R R 4.2.2, Biobase 2.58.0, GEOquery 2.66.0, limma 3.54.0, with adjusted *p*-values < 0.05 and |log_2_FC| > 1.5 as thresholds. Only cervical cancer samples and their corresponding controls were included in the differential expression analysis and examined for common characteristics, such as tumor stage, VPH type, and histology (Table 1 and Appendix A).

Across the six studies, a total of 2955 unique differentially expressed genes (DEGs) were identified. As expected, datasets with larger sample sizes or paired tumor–control designs, particularly GSE6791, GSE9750, and GSE63514, produced the highest numbers of DEGs. A graphical overview of the DEG distribution is shown in Figure 2A, where volcano plots display upregulated (red) and downregulated (blue) genes for each dataset.

To identify the recurrent signals across studies, DEGs from each dataset were compared, revealing the overlap patterns shown in Figure 2B. This visualization indicates the number of unique and shared upregulated and downregulated genes across microarrays. Consistently with the recent literature on integrative transcriptomic workflows [32], variation in DEG counts across studies reflects differences in sample size, study design, and platform chemistry. This highlights the importance of dataset-level analysis before identifying common transcriptional changes.

### 2.2. Gene Ontology (GO) Terms and KEGG Pathway Analysis by GSE

Once the DEGs were determined for each of the microarrays, a gene ontology analysis was carried out to identify the cellular, molecular, and biological mechanisms, as well as the main signaling pathways, transcription factors, and miRNAs that are most representative and shared across the different microarrays. Within the results, we can observe processes such as cell death (GO Biological), regulation of DNA replication (Pathways), protein binding regulation processes (GO Molecular), and the JUN transcription factor (Transcription factors), standing out as elements that are involved and shared in the six microarrays selected for this study (Figure 3).

However, we also observed that other elements were shared in microarrays 3, 4 and 5, such as the signaling pathways associated with IL-17 (Pathways), viral interactions related to cytokine receptors (Pathways), epidermis development (GO Biological), different extracellular elements such as vesicles or exosomes (GO Cellular), transcription factors like RELA, HDAC, and FOXM1, and miRNAs such as hsa-miR-215-5p, hsa-miR-26b-5p, and hsa-miR-335-5p standing out as elements involved in the development of CC in at least three of the six microarrays (Figure 3). Exploring transcription factors SP1, ELF3, E2F1, TP53, RELA, HDAC, and FOXM1 and their relationship with cervical cancer can reveal essential function in cancer progression. Importantly, the cellular component analysis was linked to cell death, protein binding, DNA replication, JUN transcription factor, and several miRNAs (miR-192-5p, miR-193b, and miR-215-5p) being the most representative changes among the six microarrays.

### 2.3. Microarray Analysis Integration and Common Gene Determination

To derive a high-confidence gene signature and reduce inter-dataset variability, we adopted an intersection-based integration strategy, retaining only genes that were consistently dysregulated in more than four of the six datasets (Table 2). This approach is widely used in integrative genomics to identify reproducible molecular alterations and parallels the strategy applied by Farrim et al. (2024) [32], who selected genes shared across more than four transcriptomic datasets to ensure robustness and reduce study-specific bias.

A similar tendency was observed among the 134 selected DEGs: 62 and 72 genes were upregulated and downregulated, respectively, in cervical cancer versus the control samples (Figure 4).

We selected the most prominently altered genes: 11 were upregulated coding genes (*KIF4A*, *MCM5*, *RFC4*, *PLOD2*, *MMP12*, *PRC1*, *TOP2A*, *MCM2*, *RAD51AP1*, *KIF20A*, and *AIM2*) and 14 were downregulated genes (*CXCL14*, *KRT1*, *KRT13*, *MAL*, *SPINK5*, *EMP1*, *CRISP3*, *ALOX12*, *CRNN*, *SPRR3*, *PPP1R3C*, *IVL*, *CFD*, and *CRCT1*), which were possibly involved in the cause and/or consequence, as well as biomarkers for cervical cancer. It should be noted that data from gene ontology and DEGs were different: while in the former, the TFs (*SP1*, *ELF3*, *E2F1*, *P53*, *RELA*, *HDAC*, and *FOXM1*) were the most relevant, they did not appear as DEGs. However, in later analysis, different types of genes (*CXCL14*, *KRT1*, *KRT13*, *MAL*, *SPINK5*, *EMP1*, *KIF4A*, *MCM5*, *RFC4*, *PLOD2*, *CRISP3*, *MMP12*, *ALOX12*, *PRC1*, *TOP2A*, *MCM2*, *CRNN*, *RAD51AP1*, *SPRR3*, *KIF20A*, *PPP1R3C*, *AIM2*, *IVL*, *CFD*, and *CRCT1*) were found to be more prominently dysregulated (Figure 4).

### 2.4. Protein–Protein Interaction Network Analyses

Once we determined the 134 DEGs shared between the microarrays and their expression trends, we continued our study by performing a protein–protein interaction analysis, generating an interaction network determining 10 DEGs (KIF4A, NUSAP1, BUB1B, CEP55, DLGAP5, NCAPG, CDK1, MELK, KIF11, and KIF20A) of the greatest importance (hub proteins), based on their evidence of interaction as well as the number of interactions they presented (Figure 5). Additionally, we observed that these 10 DEGs were characterized by being consistently overexpressed in their respective microarrays. Expression and interaction analysis identified the different genes associated with cervical cancer; therefore, their importance should be integrated. Interestingly, the proteins (hub) that participated in the protein–protein interaction network were not the first DEGs; however, they could be linked to TFs in GO analysis (Figure 5).

### 2.5. Functional Enriched Process for Hub Proteins

After the protein–protein interaction network was generated and we had obtained the 10 most relevant hub proteins in the network, we performed an ontology and signaling pathway analysis on these 10 elements, finding that CDK1 (cyclin-dependent kinase 1), Human kinesin family member 11 (KIF11), kinesin superfamily 4A (KIF4A), kinesin superfamily 20A (KIF20A), and Nucleolar and spindle-associated protein 1 (NUSAP1) were the five elements with the most significant participation in different processes related to the cytoskeleton, ATP synthesis, and cell-cycle regulation (Figure 6).

Additionally, their participation in GO, DEGs, and hub analysis was also remarked upon, suggesting an implication of the cause and/or consequence of cancer cervical. Cdk1 is known as a transcriptional target of E2F1 [33] and Sp1 proteins [34] which is responsible for G1/S progression [33,34], and its overexpression has been stated in cervical cancer [35]. CDK1 upregulation at the mRNA and protein level through HPV-E6 p53 modulation has been suggested [36,37]. On the other hand, the KIF11 protein plays a vital role in cell-cycle regulation and it has been implicated in the tumorigenesis and progression of various cancers, except in cervical cancer [34]. However, Kinesin Family Member 4A (KIF4A), a member of the kinesin 4 subfamily of kinesin-related proteins, serves an important role in cell division, and its overexpression has been shown in cervical cancer [38,39].

### 2.6. Hub Proteins and miRNAs Interaction Network Analyses

To determine the possible association between miRNAs and hub proteins, we used the online tool miRNet. A total of 149 miRNAs that have been experimentally reported to have had some type of interaction with at least two of the hub proteins were analyzed. For subsequent studies, only 31 miRNAs that presented an interaction with at least five or more of the hub proteins, were selected (Figure 7A,B).

Additionally, miRNA expression in cervical cancer and mRNA regulation were searched in previously published papers (Table 3). Based on previous reports of miRNA expression, we separated them into three categories: (1) miRNAs with an experimentally defined expression profile, (2) miRNAs with dual expression that were upregulated and downregulated, and (3) miRNAs with no previous experimental reports. Interestingly, the mRNAs of the hub genes were shown to be regulated by miR-103-3p, miR-107, miR-124-3p, miR-129-2-3p, miR-147a, miR-16-5p, miR-205-5p, and miR-34a-5p, suggesting gene interrelation acting as cause and/or consequence of cervical cancer. Additionally, miR-1-3p, miR-126-3p, miR-449-5p, and miR-195-5p were displayed, regulating 9 of the 10 hub genes, except for KIF20A, MELK, and KIF11, respectively.

Furthermore, miR-10b-5p, miR-23b-3p, miR-449a, and miR-130a-3p were found to modulate 4, 4, 6 and 7 of the 10 hub genes, respectively. On the other hand, miR-155-5p and miR182-5p were shown to regulate 6 of the 10 hub genes, while miR-200b-3p, miR-138-5p, miR-192-5p, miR-193b-3p, miR-203a-3p, miR-210-3p, miR-214-3p, miR-34c-5p, and let-7b-5p were found to modulate 5 of the 10 hub genes, Figure 7A,B. Regulation of the hub genes through upregulated or downregulated microRNAs has previously been shown. In this instance, microRNAs, miR-103-3p, miR-205-5p, miR-130a-3p, miR-192-5p, and miR-210-3p have been found to be overexpressed, while miR-107, miR-124-3p, miR-147a, miR-16-5p, miR-34a-5p, miR-34c-5p, miR-126-3p, miR-10b-5p, miR-23b-3p, miR-200b-3p, miR-138-5p, miR-203a-3p, miR-214-3p, and let-7b-5p expression has been revealed to be reduced. On the other hand, the expression of miR-195-5p, miR-155-5p, miR-193b-3p, and miR-449a has been ambiguous, with upregulation or downregulation in cervical cancer; and the expression of miR-129-2-3p and miR-1-3p, miR-449b-5p, and miR-522-5p in cervical cancer is unknown, Table 3.

### 2.7. miRNAs Functional Enriched Processes

Functional enriched processes for miRNAs (Figure 8) revealed nuclear organization, including nucleoplasm, nuclear lumen, and nuclear chromatin, suggesting gene regulation at the organelle shape and dynamic organization levels.

Nuclear organization is as dependent on protein binding as it can be on enzyme binding, DNA-binding transcription factor activity, RNA polymerase II-specificity, and DNA binding. This type of regulation could be linked to the TGF-beta signaling pathway and the cell cycle. Interestingly, in the synapse, microRNAs in cancer, and DNA-binding transcription factor activity, RNA polymerase II-specific processes were regulated by 12, 9, and 8 miRNAs, respectively, which were ranked as the most representative among all processes. DNA-binding transcription factor activity and RNA polymerase II-specific processes were linked to nuclear chromatin by miR-130a-3p, miR-203a-3p, miR-23b-3p, and miR-374a-5p, Figure 8. Furthermore, TGF-beta signaling pathway, DNA-binding transcription factor activity, RNA polymerase II-specific, and nuclear chromatin processes were regulated by miR-130a-3p and miR-23b-3p. Interestingly, microRNAs in cancer were linked to nervous system development and enzyme binding by miR-34c-5p, miR-449a, and miR-449b-5p. In agreement, the activation of nervous-system-related genes in cervical cancer has been previously reported [77].

## 3. Discussion

The bioinformatics analyses performed on six cervical cancer expression microarray studies showed 1471 and 1492 upregulated and downregulated coding genes, respectively. Across the different platforms’ analysis, a total of 134 coding genes presented similar expression in at least four of the six microarrays analyzed. From 134 common genes, 62 and 72 were upregulated and downregulated, respectively. The GO analysis showed several associations with cell death processes, regulation of DNA replication, protein-binding regulation, and transcription factors as JUN, SP1, ELF3, E2F1, P53, RELA, HDAC, and FOXM1, which have been reported to contribute to the promotion of cervical cancer, which is also regulated by oncomiRs and tumor-suppressing microRNAs [78,79,80], which are also associated with DEGs in this work. In agreement with the results found here, the impact of this microRNA deregulation on cervical cancer has been previously stated. hsa-miR-26b-5p has been reported to be increased in cervical cancer [81,82], and upregulation of hsa-miR-215-5p has been related to poor survival [83]. However, hsa-miR-335-5p, found to be overexpressed in our analysis, has been previously reported as an anti-oncomiRNA [84]; therefore, this preliminary analysis may be inconclusive and the study population differences such as ethnicity, HPV infection, and advancement of the disease may be playing a role in this discrepancy. Eleven coding genes were upregulated (*KIF4A*, *KIF20A*, *MCM2*, *MCM5*, *TOP2A*, *RFC4*, *RAD51AP1*, *PLOD2*, *MMP12*, *PRC1*, and *AIM2*) and 14 were downregulated (CXCL14, *KRT1*, *KRT13*, *MAL*, *SPINK5*, *EMP1*, *CRISP3*, *ALOX12*, *CRNN*, *SPRR3*, *PPP1R3C*, *IVL*, *CFD*, and *CRCT1*); these genes were found to be differentially expressed between cervical cancer and the controls. It was previously shown that several proteins participate in a cellular program to enhance DNA replication by an increase in replicative helicase proteins (MCM2, MCM4, MCM5, MCM6, and MCM10), DNA polymerases (PLOA1/E2/E3/Q), and cytokinesis through motor proteins (KIF11, KIF14, KIF4A, and PRC1) in cervical cancer progression [39]. KIF4A is dominantly localized in the nuclear matrix and is associated with chromosomes during mitosis [85] and mediates cytokinesis during cervical cancer progression [39]. Another Kinesin family member, KIF20A, has been correlated with HPV infection, clinical stage, tumor recurrence, lymphovascular space involvement, pelvic lymph node metastasis, and poor outcome in early-stage cervical squamous cell carcinoma patients [86]. Additionally, it should be noted that in our work, the genes from GO and protein–protein interaction analyses were different. The upregulated coding genes were mainly related to cytokinesis, DNA repair, and replication. Interestingly, their miRNAs regulators are downregulated. The expression of miR-107 [41,42], miR-124-3p [44,53], miR-126-3p [45,46], miR-138-5p [49,50], miR-16-5p [53,54], miR-200b-3p [61,62], miR-203a-3p [63], miR-214-3p, miR-23b-3p, miR-26a-3p, miR-34a-5p, and miR-34c-5p have been previously reported as diminished, similarly to our data. Interestingly, the same microRNAs that regulate the DEGs modulate the genes resulting from protein–protein interaction network analyses (KIF4A, NUSAP1, BUB1B, CEP55, DLGAP5, NCAPG, CDK1, MELK, KIF11, and KIF20A); these denominated hub proteins are based on their evidence of interaction, as well as the number of interactions they present. Hub proteins have been implicated in disease progression, ensuring constant tumorigenesis-related proteins. Likewise, NUSAP1 is a crucial mitotic regulator that binds to microtubules and mediates their attachment to chromosomes, ensuring the accurate distribution of genetic material to the two daughter cells, and it is associated with cervical cancer progression [87]. HPV 18 E6/E7 promoted maternal embryonic leucine zipper kinase (MELK) expression by activating E2F1 [30] and DLGAP5 stabilized E2F1 through its binding, preventing the ubiquitination of E2F1 via USP11 [88]. E2F1 binds to the promoter of CDK1 [30], NCAPG [89], and TOP2A, promoting its transcription [90], which is potentiated by the expression of E7 [91]. CDK1 is known as a transcriptional target of Sp1 [31], which is responsible for G1/S progression [33,34]. HPV-E6 upregulates CDK1 at mRNA and protein [36], suggesting that CDK1 modulation is p53-dependent [37]. p53 activates the transcription p21 a Cdk inhibitor. CDK1 binds p21 with lower affinity than Cdk2, abrogating the postmitotic checkpoint in E6-expressing cells, favoring cervical cancer development through the induction of polyploidy [92]. The level of the protein p53 could influence CDK1 expression. It has been reported that miR-92a-1-5p downregulates the expression of TP53 [93]. The Cyclin B1/CDK1- complex induces the phosphorylation of PRC1 [94], a protein present at high levels during the S and G2/M phases of the cell cycle. MMP12 expression is regulated positively by MTA2 via AP1 through phosphorylation of the pathway ASK1/MEK3/p38/YB1, leading to tumor cell metastasis [95]. JUN has been reported to increase in cancer versus CIN and normal tissue [96]. The expression of phosphorylated c-Jun, c-Fos, and ERK1/2, a key factor of the ERK signaling pathway, was increased in the progressive lesions of the cervix [97]. Several genes have been reported as being regulated by miRNAs, while MELK is regulated by miR-107 [41,42], miR-124-3p [44,53], miR-16-5p [53,54], miR-200b-3p [61,62], miR-203a-3p [63], miR-214-3p, miR-23b-3p, miR-26a-3p, and miR-34a-5p. DLGAP5 is modulated by miR-107 [41,42,70], miR-124-3p [44,53], miR-126-3p [45,46], miR-16-5p [53,54], miR-200b-3p [61,62], miR-23b-3p, miR-26a-3p, miR-34a-5p, and miR-34c-5p; it should be noted that there is a difference of two miRNAs (miR-203a-3p, miR-214-3p) and three miRNAs (miR-126-3p, miR-200b-3p and miR-34c-5p), respectively, between these two coding genes. E2F1 downregulates miR-107 through its binding to the promoter of miR-107, inducing transcriptional repression [98]. On the other hand, E2F1 is downregulated by miR-16-5p [99] and miR-34c-5p [84]. This study used multiple microarray datasets of patients diagnosed with cervical cancer, but some data, such as associated comorbidities, duration of illness, or presence and related HPV, could not be fully represented due to limitations in the GEO database or the data reported by microarray authors. Furthermore, microarray technology has limitations, such as platform-specific probe differences, incomplete transcript coverage, and several variations in metadata, which can limit the interpretation of associations with disease progression or comorbidities not reported from samples. Meanwhile, PPI analysis depends on the existing experimental data. Additionally, protein interaction databases may not include all possible interactions, especially for poorly understood proteins or pathways. Similarly, the identification of miRNAs related to DEGs in cervical cancer was performed using two platforms, such as miRNet and RNADisease, but they rely on known interactions, which can cause discrepancies between databases. Additionally, the bioinformatics analysis of biological pathways, molecular functions, transcription factors, and miRNA interactions are based on the current definitions and databases, which may be incomplete or outdated. Remarkably, we aimed to integrate multiple microarray datasets to enable a broad identification of genes with potential biomedical importance in cervical cancer, including samples from various HPV genotypes (primarily HPV16), different tumor stages (IB–IIB), and diverse squamous histological subtypes introducing clinical variability that reflects the biological diversity of the disease and enhances the translational relevance of the findings. Furthermore, the processes found herein were linked to different genes; therefore, it is imperative to integrate the processes, systems, and gene regulation. Unraveling the coding and non-coding RNAs regulated during cervical cancer is an important factor in achieving new treatments and biomarkers.

## 4. Materials and Methods

### 4.1. General Diagram

Gene expression datasets for cervical cancer were retrieved from the GEO database, and differentially expressed genes (DEGs) were identified by using GEO2R R 4.2.2, Biobase 2.58.0, GEOquery 2.66.0, limma 3.54.0 (|logFC| > 1.5, adjusted *p* < 0.05). Curated gene lists (based on HUGO nomenclature and Venn analysis) were used to construct a protein–protein interaction (PPI) network in STRING v12, with the hub genes determined through cytoHubba. Gene ontology (GO) and pathway enrichment analyses were performed using EnrichR. Hub gene–associated miRNAs were predicted with miRNet 2.0, disease-specific associations were retrieved from the RNA Disease database, and functional miRNA analyses were conducted in miRPathDB 2.0. The integrated analysis identified key genes, miRNAs, and pathways associated with cervical cancer (Figure 1).

### 4.2. Gene Expression Profile Data Collection

Microarray samples from healthy patients and patients with cervical cancer from 6 different tissue experiments of gene expression were selected from the NCBI Gene Expression Omnibus repository (www.ncbi.nlm.nih.gov/geo/, (accessed on 13 June 2025)) (Table 1). Raw data can be revised in Appendix A.

### 4.3. Identification of Differentially Expressed Genes

Differentially expressed genes (DEGs) were identified independently for each microarray by using the GEO2R bioinformatics tool. This strategy helps to avoid potential biases associated with the batch effect, which can arise when combining heterogeneous datasets in a single analysis.

This ensured that comparisons between the different microarrays were performed only with the most relevant DEGs in each case, in a robust manner and without introducing external technical variability. During the analysis, only samples corresponding to controls and cases with cervical cancer were selected.

Only the genes with a log FC (fold-change) > |1.5| and adjusted *p*-value < 0.05 were considered to be statistically significant in the comparisons of cervical cancer and the controls, as recommended by DJ McCarthy and GK Smyth in 2009 for the analysis of biological data [100].

The adjusted *p*-value (adj. *p*), using the Benjamini–Hochberg (BHC) procedure, was applied to limit the false discovery rate (BHC method) [101]. The obtained upregulated and downregulated DEGs were classified into the cervical cancer and control groups, and the shared information from each microarray was identified with the help of Venn diagrams created with an online platform (http://bioinformatics.psb.ugent.be/webtools/Venn/ (accessed on 16 December 2025)) [102]. The shared information was organized in lists and curated by eliminating synonymous gene names with the HUGO Gene Nomenclature Committee platform (https://www.genenames.org/) [103].

### 4.4. PPI Network Construction

The DEG lists selected for the CC and controls association were uploaded into the STRING platform (STRING; v12.0; https://string-db.org/) to create protein–protein interaction networks (PPI). The parameters for constructing the DEG CC/Control network on STRING included experimental information, database curation, and co-expression evidence to reduce the rate of false-positive results (medium confidence, 0.400) [104]. Only the nodes that had at least two connections on the produced network were kept for analysis. The hub proteins within the network were determined using the maximal clique centrality method (MCC) parameter, available in the cytoHubba app for Cystoscope (v3.10.1) [105,106]. To determine the proteins of high biological value within the PPI network groups (hub proteins), the hub option from the EnrichR platform (https://maayanlab.cloud/Enrichr/ (accessed on 16 December 2025)) was used. The Expression2Kinases program (https://www.maayanlab.net/X2K/ (accessed on 16 December 2025)) [107] was used to identify the regulatory proteins (mainly transcription factors (TFs) and kinases) involved in important signaling pathways that potentially regulate a PPI network based on the gene list submitted [108].

### 4.5. DEG Annotation and Functional Analyses

The biological value of the PPI, derived from the DEGs Cervical Cancer/Controls and for the hub proteins, was determined through an enrichment analysis that included KEGG pathways, transcription factors (TFs), dbGap genotype/phenotypes, biological processes (BP), molecular functions (MF), and cellular components (CC) with the online EnrichR platform (https://maayanlab.cloud/Enrichr/ (accessed on 16 December 2025)) [108]. Only ontology annotation terms with adj. *p* < 0.05 were considered significant.

### 4.6. Computational Identification of miRNAs Associated with Cervical Cancer

The miRNAs associated with the hub genes of the DEGs-Cervical Cancer/Controls network were determined by using the miRNet platform (https://www.mirnet.ca/ (accessed on 16 December 2025)) [109]. This database contains information on interactions between miRNAs and their target gene, incorporating various database sources such as miRTarBase v8.0, TarBase v8.0, and miRecords v1.0 [110,111]. In the network analysis, the miRNAs that had at least two interactions with hub genes or were reported for cervical cancer development were kept for the subsequent analysis on the RNA-Disease platform (http://www.rnadisease.org/) [112]. On this last platform, the search for miRNAs of interest was carried out to collect experimental and computational evidence related to their expression in cervical cancer. Only miRNAs with experimental evidence were manually analyzed in detail.

### 4.7. Functional Analysis of miRNAs in miRPathDB

Functional analysis of miRNAs was performed using the miRPathDB V4.0 [113], a bioinformatics tool designed for the annotation and functional analysis of miRNAs that collects information that is validated in the literature, thus allowing us to discover some of the biological implications and roles of miRNAs. For our study, only terms with an Adj. *p* < 0.05 were considered to be significant.

## 5. Conclusions

The integrative transcriptomic and network-based analysis performed in this study provides a robust and reproducible molecular framework to understand the regulatory landscape of cervical cancer. Combining six independent GEO datasets, encompassing diverse populations and experimental platforms, we overcame cohort-specific variability and minimized batch effects, achieving a consensus gene signature that reflects biologically consistent transcriptional alterations. The hub genes KIF4A, NUSAP1, BUB1B, CEP55, DLGAP5, NCAPG, CDK1, MELK, KIF11, and KIF20A converge on pathways that are essential for mitosis, spindle formation, and cell-cycle progression, all of which are recognized hallmarks of cancer development. In parallel, the integration of miRNA–mRNA regulatory networks revealed a coordinated pattern of post-transcriptional dysregulation. miR-107, miR-124-3p, miR-16-5p, miR-126-3p, and let-7b-5p are inversely expressed to hub genes, supporting a mechanistic model that contributes to the persistent activation of mitotic and proliferative pathways, amplifying tumor aggressiveness. It should be noted that combining data from different laboratories, technological platforms, and protocols leads to a high risk of strong batch effects. These effects can mask biological differences or generate spurious signals of differential expression. The integration, consisting of very diverse samples, should be interpreted with caution and merely as a general expression signature, not as a precise result for a specific population. Our approach was adopted to minimize potential cohort-specific biases and highlight only the transcriptional changes that were consistently or globally observed in independent analyses, identifying only the common expression pattern related to the disease, rather than effects associated with specific populations or contexts. Despite strong statistical support, the results still lack functional and experimental validation. However, these findings provide a system-level perspective, linking transcriptional deregulation with disrupted post-transcriptional control and thereby strengthening the molecular basis for future biomarker validation and therapeutic targeting in cervical cancer.

## Figures and Tables

**Figure 1 ijms-27-00258-f001:**
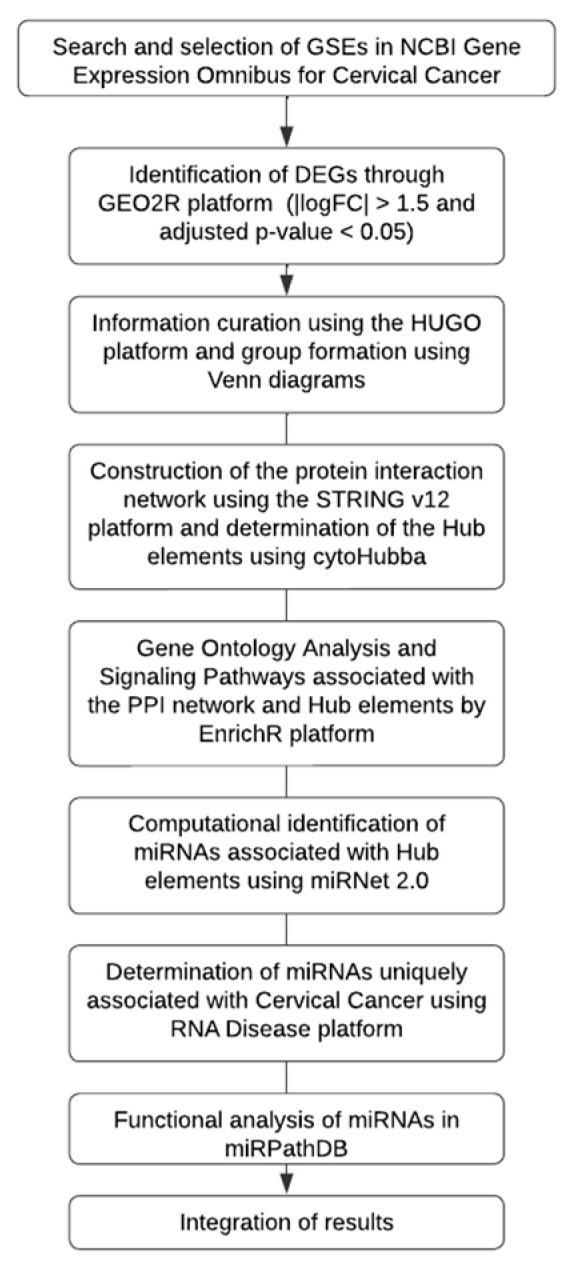
Workflow of the bioinformatics analysis. Identification of DEGs from GEO datasets, construction of PPI networks, enrichment analysis, and integration of miRNA functional data associated with cervical cancer. Each microarray was analyzed independently, and a common pattern of gene expression was selected for the subsequent analysis following this consideration (adj. *p*  <  0.05, |logFC|  >  1.5).

**Figure 2 ijms-27-00258-f002:**
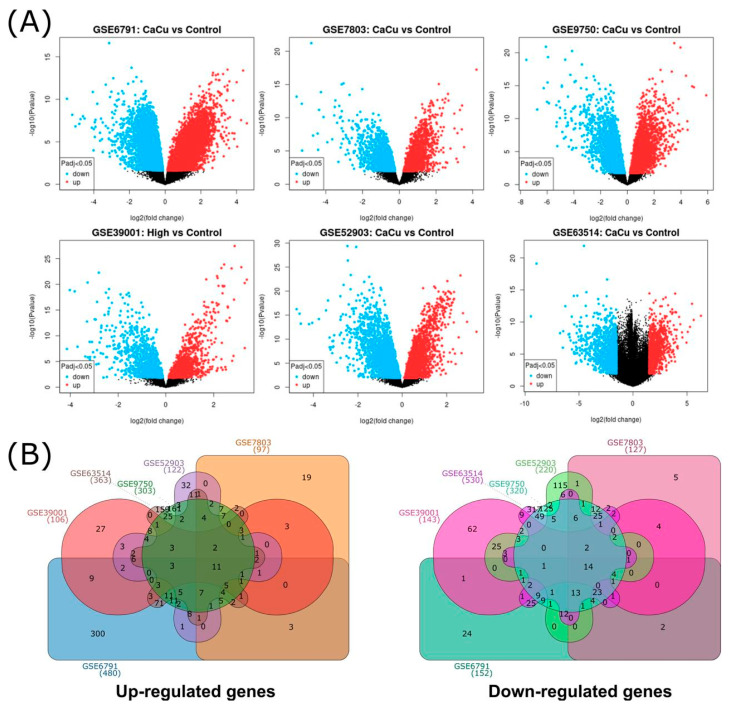
General distribution of DEGs by microarray. (**A**) Volcano plots from 6 microarray analyses, blue and red dots represent upregulated and downregulated genes, respectively. The black dots represent genes with basal expression. (**B**) Common upregulated and downregulated differentially expressed genes from cervical cancer visualized through a Venn diagram, (GSE39001 (106 up and 143 down DEGs), GSE52903 (122 up and 220 down DEGs), GSE63514 (363 up and 530 down DEGs), GSE6791 (480 up and 152 down DEGs), GSE7803 (97 up and 127 down DEGs), and GSE9750 (303 up and 320 down DEGs)). The numbers indicate the quantity of genes that were shared between microarrays.

**Figure 3 ijms-27-00258-f003:**
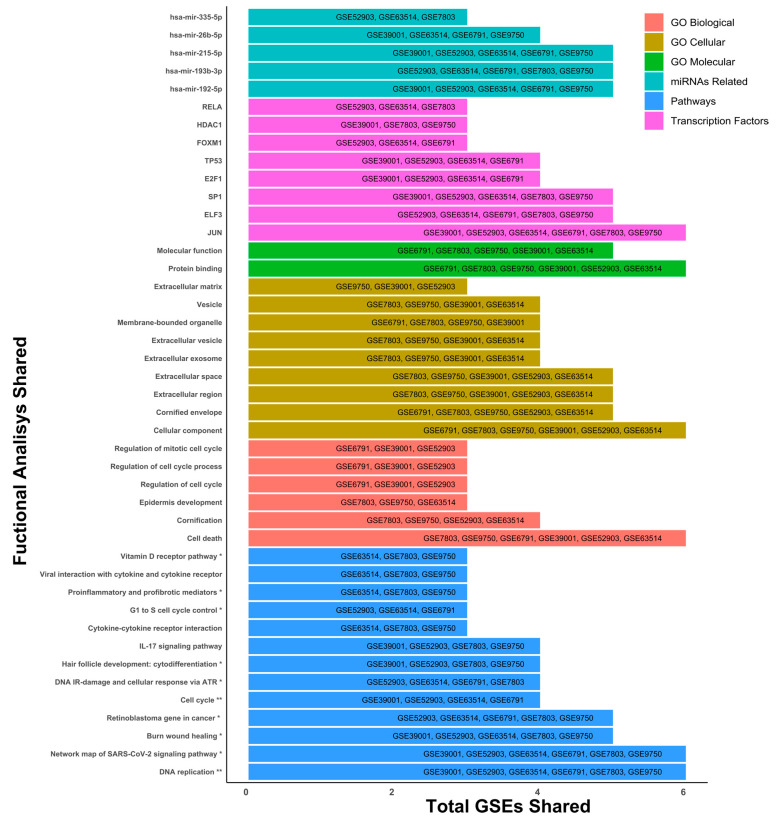
General ontology analysis. Gene ontology was determined with 197 tissue samples with cervical cancer and 95 controls. Differentially expressed genes with a threshold of 1.5 and *p*-adjusted < 0.05 were used for ontology analysis. The pathway analysis was performed using the KEGG and Wiki Pathways databases; * is significant in only one database and ** is significant in both databases.

**Figure 4 ijms-27-00258-f004:**
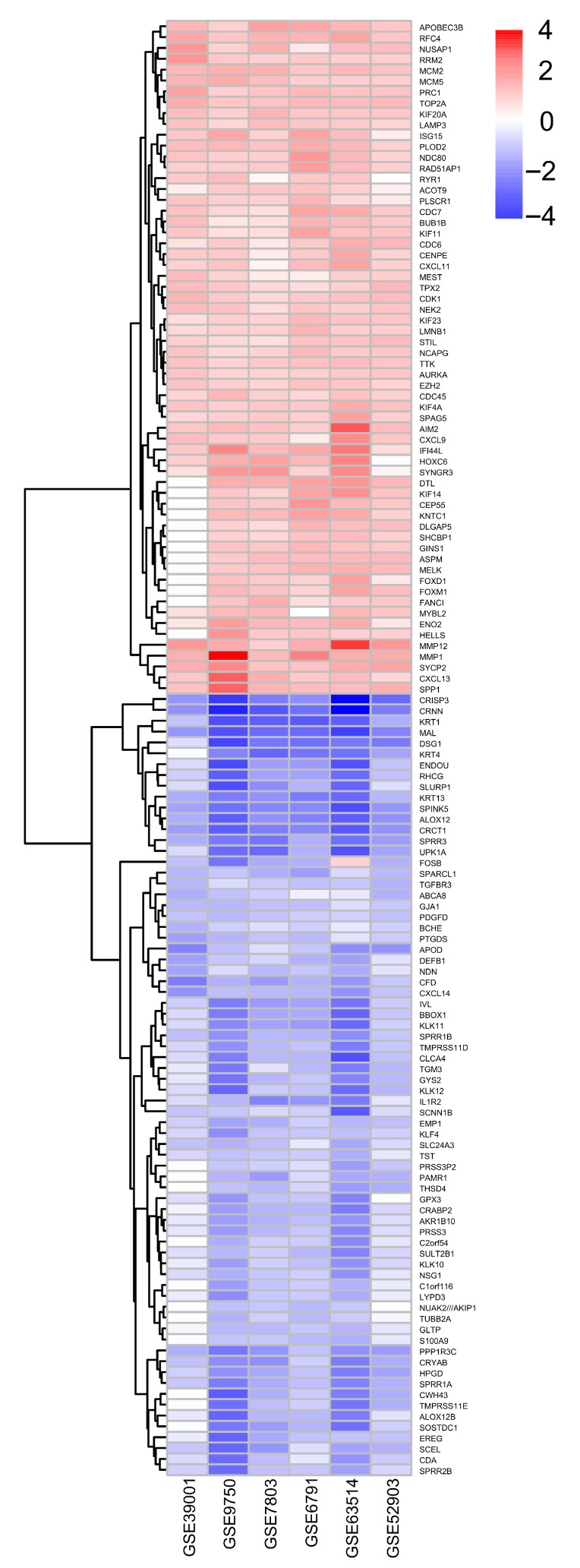
Differentially expressed genes between microarrays. Differentially expressed genes that were upregulated and downregulated were grouped based on their fold change of 1.5 and *p*-adjusted < 0.05.

**Figure 5 ijms-27-00258-f005:**
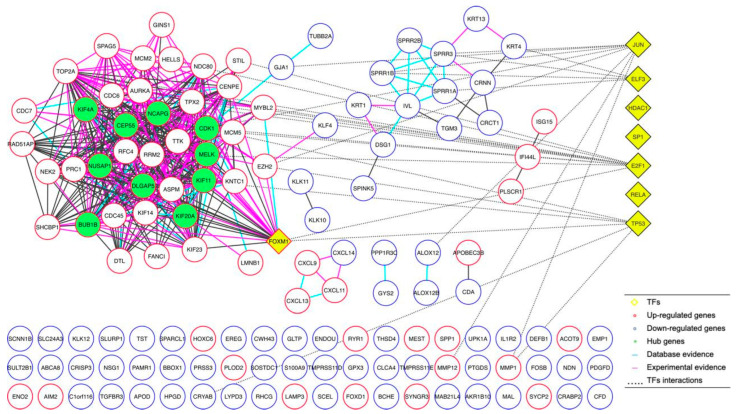
Protein–protein interaction network analyses. The analysis of protein–protein interaction using 134 DEGs shared between the microarrays, generating an interaction network.

**Figure 6 ijms-27-00258-f006:**
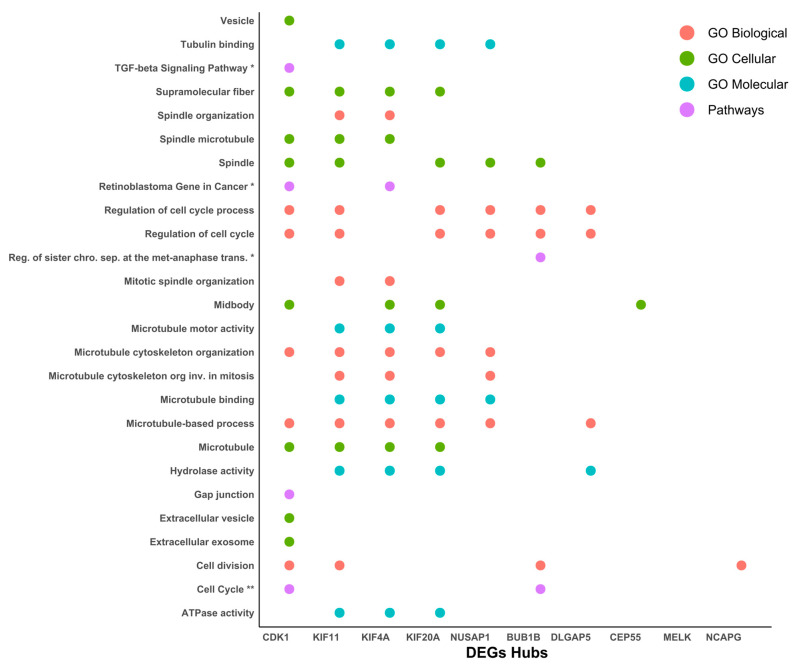
Functional enriched process for hub proteins. Ontology and signaling pathway analysis were performed on the protein–protein interaction network. The pathway analysis was performed using the KEGG and Wiki Pathways databases; * is significant in only one database and ** is significant in both databases.

**Figure 7 ijms-27-00258-f007:**
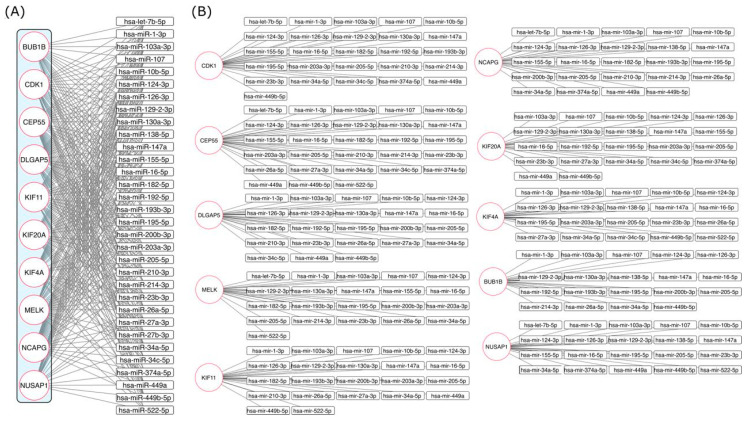
Interaction network between mRNAs of hub proteins and miRNAs. (**A**) miRNAs and mRNAs of hub protein interactions were determined using the online tool miRNet. (**B**) miRNAs and mRNAs of hub protein interactions were separated and amplified to clarify their association.

**Figure 8 ijms-27-00258-f008:**
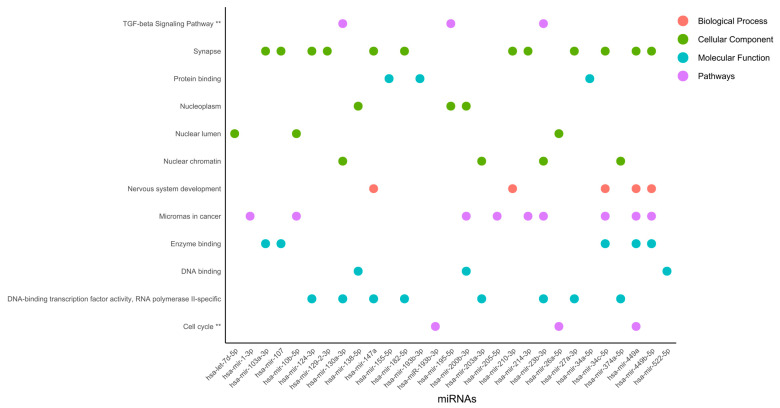
Functional enriched processes regulated by miRNAs. miRNAs were classified into 4 categories of processes regulated by miRNAs involved in cervical cancer. The pathway analysis was performed using the KEGG and Wiki Pathways databases; ** is significant in both databases.

**Table 1 ijms-27-00258-t001:** General information on expression sets and platforms from cervical cancer and healthy control samples. SCC: Squamous Cell Carcinoma.

Microarray (GSE)	Platform	Dominant VPH Type	Dominant Tumor Stage	Dominant FIGO Staging	Cervical Cancer Samples
GSE39001	GPL201	16	IB1	SCC	43
GSE9750	GPL96	16	IIB	SCC	17
GSE7803	GPL96	16	Not reported	SCC	21
GSE6791	GPL570	16	IB	SCC	28
GSE63514	GPL570	16	Not reported	SCC	28
GSE52903	GPL6244	Not reported	IB1	SCC	55

**Table 2 ijms-27-00258-t002:** Distribution of the main DEGs between GSEs.

Datasets (GSE)	Main Differentially Expressed Genes by Expression Trend
GSE39001 GSE52903 GSE63514 GSE6791 GSE7803 GSE9750	* KIF4A MCM5 RFC4 PLOD2 MMP12 PRC1 TOP2A MCM2 RAD51AP1 KIF20A AIM2 *
GSE39001 GSE52903 GSE63514 GSE6791 GSE7803 GSE9750	* CXCL14 KRT1 KRT13 MAL SPINK5 EMP1 CRISP3 ALOX12 CRNN SPRR3 PPP1R3C IVL CFD CRCT1 *
GSE39001 GSE63514 GSE6791 GSE7803 GSE9750	*CXCL13 NDC80 ISG15 IFI44L HOXC6*
GSE39001 GSE52903 GSE6791 GSE7803 GSE9750	*MMP1*
GSE39001 GSE52903 GSE63514 GSE7803 GSE9750	*CXCL9 SYCP2*
GSE39001 GSE52903 GSE63514 GSE6791 GSE9750	*CDK1 RRM2 NEK2*
GSE39001 GSE52903 GSE63514 GSE6791 GSE7803	*APOBEC3B TTK*
GSE39001 GSE52903 GSE6791 GSE7803 GSE9750	* GJA1 PTGDS SPARCL1 FOSB *
GSE39001 GSE52903 GSE63514 GSE7803 GSE9750	* CRYAB SCEL *
GSE39001 GSE52903 GSE63514 GSE6791 GSE9750	* APOD *
GSE39001 GSE52903 GSE63514 GSE6791 GSE7803	* TGFBR3 *
GSE52903 GSE63514 GSE6791 GSE7803 GSE9750	* GINS1 KNTC1 MELK ASPM SPAG5 CEP55 DTL *
GSE52903 GSE63514 GSE6791 GSE7803 GSE9750	* BBOX1 UPK1A ENDOU KLK12 DSG1 HPGD TMPRSS11D RHCG SPRR1A CLCA4 KRT4 GYS2 SPRR1B *
GSE39001 GSE6791 GSE7803 GSE9750	*PLSCR1*
GSE39001 GSE52903 GSE7803 GSE9750	*SPP1*
GSE39001 GSE63514 GSE6791 GSE9750	*EZH2 RYR1 CENPE*
GSE39001 GSE52903 GSE63514 GSE9750	*TPX2 CXCL11 MEST*
GSE39001 GSE63514 GSE6791 GSE7803	*LAMP3*
GSE39001 GSE52903 GSE63514 GSE7803	*NUSAP1*
GSE39001 GSE52903 GSE63514 GSE6791	* KIF11 AURKA CDC7 STIL BUB1B NCAPG *
GSE39001 GSE63514 GSE7803 GSE9750	* SLC24A3 *
GSE39001 GSE52903 GSE7803 GSE9750	* ABCA8 PDGFD *
GSE39001 GSE63514 GSE6791 GSE9750	* DEFB1 SCNN1B *
GSE39001 GSE52903 GSE6791 GSE9750	* BCHE *
GSE39001 GSE63514 GSE6791 GSE7803	* NDN *
GSE63514 GSE6791 GSE7803 GSE9750	*SYNGR3 ENO2 ACOT9 FOXD1*
GSE52903 GSE6791 GSE7803 GSE9750	*HELLS*
GSE52903 GSE63514 GSE7803 GSE9750	*FOXM1 MYBL2 FANCI CDC45*
GSE52903 GSE63514 GSE6791 GSE9750	*CDC6 KIF14 KIF23 SHCBP1 DLGAP5*
GSE63514 GSE6791 GSE7803 GSE9750	* GLTP AKR1B10 SPRR2B KLF4 LYPD3 PRSS3 TUBB2A CRABP2 IL1R2 C2orf54 TST ALOX12B NUAK2///AKIP1 KLK10 SOSTDC1 GPX3 SLURP1 NSG1 KLK11 SULT2B1 C1orf116 S100A9 *
GSE52903 GSE6791 GSE7803 GSE9750	* EREG *
GSE52903 GSE63514 GSE7803 GSE9750	* TMPRSS11E CWH43 PRSS3P2 PAMR1 THSD4 CDA *
GSE52903 GSE63514 GSE6791 GSE9750	* TGM3 *
GSE52903 GSE63514 GSE6791 GSE7803	*LMNB1*

Note: Upregulated genes are shown in red and downregulated genes are shown in blue.

**Table 3 ijms-27-00258-t003:** miRNAs and targets regulated in cervical cancer. — No information was found in cervical cancer, ↑ overexpressed miRNAs in cervical cancer, ↓ underexpressed in cervical cancer, ↓↑ overexpressed and underexpressed in cervical cancer.

miRNAs	Cervical Cancer	DEGs Targets	Reference
hsa-let-7b-5p	—	NUSAP1, NCAPG, MELK, CEP55, CDK1	
hsa-mir-1-3p	—	BUB1B, CDK1, CEP55, DLGAP5, KIF11, KIF4A, MELK, NCAPG, NUSAP1	
hsa-mir-103a-3p	↑	KIF20A, CEP55, CDK1, NUSAP1, DLGAP5, KIF11, NCAPG, BUB1B, KIF4A, MELK	[40]
hsa-mir-107	↓	BUB1B, CDK1, CEP55, DLGAP5, KIF11, KIF20A, KIF4A, MELK, NCAPG, NUSAP1	[41,42]
hsa-mir-10b-5p	—	CDK1, CEP55, DLGAP5, KIF11, KIF20A, KIF4A, NCAPG, NUSAP1	
hsa-mir-124-3p	↓	BUB1B, CDK1, CEP55, DLGAP5, KIF11, KIF20A, KIF4A, MELK, NCAPG, NUSAP1	[43,44]
hsa-mir-126-3p	↓	BUB1B, CDK1, CEP55, DLGAP5, KIF11, KIF20A, KIF4A, NCAPG, NUSAP1	[45,46]
hsa-mir-129-2-3p	—	BUB1B, CDK1, CEP55, DLGAP5, KIF11, KIF20A, KIF4A, MELK, NCAPG, NUSAP1	
hsa-mir-130a-3p	↑	BUB1B, CDK1, CEP55, DLGAP5, KIF11, KIF20A, MELK	[47,48]
hsa-mir-138-5p	↓	BUB1B, KIF20A, KIF4A, NCAPG, NUSAP1	[49,50]
hsa-mir-147a	—	BUB1B, CDK1, CEP55, DLGAP5, KIF11, KIF20A, KIF4A, MELK, NCAPG, NUSAP1	
hsa-mir-155-5p	↓↑	CDK1, CEP55, KIF20A, MELK, NCAPG, NUSAP1	[51,52]
hsa-mir-16-5p	↓	BUB1B, CDK1, CEP55, DLGAP5, KIF11, KIF20A, KIF4A, MELK, NCAPG, NUSAP1	[53,54]
hsa-mir-182-5p	—	CDK1, CEP55, DLGAP5, KIF11, MELK, NCAPG	
hsa-mir-192-5p	↑	BUB1B, CDK1, CEP55, DLGAP5, KIF20A	[55,56]
hsa-mir-193b-3p	↓↑	BUB1B, CDK1, KIF11, MELK, NCAPG	[57,58]
hsa-mir-195-5p	↑↓	BUB1B, CDK1, CEP55, DLGAP5, KIF11, KIF20A, KIF4A, MELK, NCAPG, NUSAP1	[59,60]
hsa-mir-200b-3p	↓	BUB1B, DLGAP5, KIF11, MELK, NCAPG	[61,62]
hsa-mir-203a-3p	↓	CDK1, CEP55, KIF11, KIF20A, KIF4A, MELK	[63]
hsa-mir-205-5p	↑	BUB1B, CDK1, CEP55, DLGAP5, KIF11, KIF20A, KIF4A, MELK, NCAPG, NUSAP1	[60,64]
hsa-mir-210-3p	↑	CDK1, CEP55, DLGAP5, KIF11, NCAPG	[45,65]
hsa-mir-214-3p	↓	BUB1B, CDK1, CEP55, MELK, NCAPG	[66,67]
hsa-mir-23b-3p	↓	CDK1, CEP55, DLGAP5, KIF20A, KIF4A, MELK, NUSAP1	[68]
hsa-mir-26a-5p	↓	BUB1B, CEP55, DLGAP5, KIF11, KIF4A, MELK, NCAPG	[69,70]
hsa-mir-27a-3p	↑	CEP55, DLGAP5, KIF11, KIF20A, KIF4A	[71,72]
hsa-mir-34a-5p	↓	BUB1B, CDK1, CEP55, DLGAP5, KIF11, KIF20A, KIF4A, MELK, NCAPG, NUSAP1	[63,73]
hsa-mir-34c-5p	↓	CDK1, CEP55, DLGAP5, KIF20A, KIF4A	[74,75]
hsa-mir-374a-5p	—	CDK1, CEP55, KIF20A, NCAPG, NUSAP1	
hsa-mir-449a	↑↓	CDK1, CEP55, DLGAP5, KIF11, KIF20A, NCAPG, NUSAP1	[73,76]
hsa-mir-449b-5p	—	BUB1B, CDK1, CEP55, DLGAP5, KIF11, KIF20A, KIF4A, NCAPG, NUSAP1	
hsa-mir-522-5p	—	CEP55, KIF11, KIF4A, MELK, NUSAP1	

## Data Availability

The original contributions presented in this study are included in the article/Appendix A. Further inquiries can be directed to the corresponding author.

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
