# Peer review of "Differentially Expressed Genes Associated with the Development of Cervical Cancer"

_ijms, 2025, doi:10.3390/ijms27010258_

Round 1

Reviewer 1 Report

Comments and Suggestions for Authors

The work of Alvarado-Camacho et al. analyzes public transcriptomic data related to cervical cancer, with the goal of identifying potential biomarkers and therapeutic targets. The introduction is well written and provides a clear overview, particularly the section on miRNAs. However, the manuscript in its current form suffers from issues in Figure clarity, writing flow, bioinformatics practices, and overall design of the experiments. The description of the datasets and experimental setup is insufficient; it is notably unclear how gene lists from different datasets were integrated; especially without distinguishing up and down -regulated genes. Figures are difficult to read, and miss details in captions. The results section is often overly descriptive and lacks interpretative flow. Furthermore, the functional enrichment analyses do not seem to follow standard bioinformatic practices. The manuscript also includes numerous grammatical errors, unclear sentences, and poorly describe methods; making a thorough peer review challenging. Overall, while the topic and dataset choice are highly relevant, the manuscript requires substantial revision in writing, Figure presentation, and analytical rigor before I would recommend it for publication in International Journal of Molecular Sciences. Please see below for my major and minor points.

Major points

Line128-138: This is an important part of the introduction as it should clearly state the objective of the study. Please clarify the overall goal: are the authors aiming to identify genes that are more highly expressed in cervical cancer compared to healthy controls, or to determine whether these genes are regulated by specific miRNAs? I would also kindly suggest including Figure 8 earlier in the manuscript as it nicely summarizes the workflow and would help readers understand the overall study aim.

Figure1: The Venn diagram is difficult to interpret as it combines DEGs without distinguishing up and down -regulated genes. Some datasets may overlap due to genes significantly deregulated in the opposite direction. I would recommend separating up and down -regulated genes into two Venn diagrams. Furthermore, using a more classic round/oval Venn layout would also improve readability.

Line151-156: I do not fully agree with the authors statement that higher number of samples provides “more accurate results” or leads to a higher number of DEGs. First, Table 1 does not support this claim, as GSE63514 includes only 52 samples (24 controls and 28 cancer), not 128, yet it reports many DEGs. Second, in cancer studies, sample quality, the choice of controls, and overall tissue heterogeneity (between cancer types, and healthy controls) have a lot of impact on DEG detection.

Figure2: Please increase font size and overall Figure quality. It is also not clear how the enrichment analysis was performed; notably, what input gene list was used?, which significance threshold were applied? These details should be mention in Result section and/or Figure caption.
Also, please explain the meaning of the asterisks shown next to some terms. Finally, what the ‘Transcription Factors’ corresponds to here? Method section mentioned that the enrichment analyses perform using enrichR, but which database has been used for ‘TF’?

Another important point is that combining up and down -regulated genes in the same analysis may not be appropriate: please support this in the text, or consider separating up and down regulated genes in response to CC.

Line182-248: The result section reads more like a Discussion; there is excessive detail and interpretation, rather than concise reports of findings. The detailed functional descriptions of each TF may not be necessary, nor included here in the Result section. Please make this section more succinct, focusing on presenting key results only, and move biological interpretation to the Discussion.

Furthermore, adding a short concluding statement summarizing the main takeway message would help for clarity and flow.

The exact same comments apply for Line268-316.

Figure3: Please increase font size and overall Figure quality. What does the color intensity correspond to? What type of values does this represent? Please add detail in Figure caption.

Line258: How does the “prominently altered genes” have been selected?

Figure4: Please increase font size and overall Figure quality.

Figure5: Please increase font size and overall Figure quality. Furthermore, it is unclear how the functional enrichment analysis was performed. Does the ten genes shown on the x-axis were used as input? What does each dot represent?; Does it indicate that the gene is included in a given ontology term? The figure and caption should clearly explain these points. The same clarification is also needed for Figure 7.

Figure6: Please increase font size and overall Figure quality.

Line418-420: The three categories used by the authors to classify miRNAs are not clearly defined; please describe in more detail, notably, what is meant by “define profile”?

The Discussion section cannot be properly evaluated until the major issues outlined above are addressed.

Minor points

Line145: Which data sheet? I was not able to access it.

Table1: Please provide more detail on the sample if possible (ie. age of individuals, cancer stage, treatment, survival rate; any more information upon availabilities would be relevant to include here).

Line326-328: Are the Hub genes DE, or not? It is not clear what “not the first DEGs” means here.

Typos

Line48: “associated to” seems incorrect; “in response to”?

Line67-69: This sentence is not clear notably “that change transitory normal to immortal cell functions” please rephrase for clarity.

Line78-80: What “is presented”? Please rephrase for clarity.

Line129-133: Repetitive, please rephrase.

Line178: What “can be advised”? Please rephrase for clarity.

Line183-185: This sentence is not clear, please rephrase.

Line203-205: “International Federation of Gynecology and Obstetrics”?

Line249: Title missing for 2.1.2

Line264: “they did not appear as DEGs” ; miss a word

Line323: Figure4, and not Figure3.

Line382-406: Part “Functional enriched process for HUB proteins” is duplicated.

Line469: “whit”

Line479: Please rephrase.

Author Response

Dear reviewer, I tried to respond to all suggestions and observations.

The work of Alvarado-Camacho et al. analyzes public transcriptomic data related to cervical cancer, with the goal of identifying potential biomarkers and therapeutic targets. The introduction is well written and provides a clear overview, particularly the section on miRNAs. However, the manuscript in its current form suffers from issues in Figure clarity, writing flow, bioinformatics practices, and overall design of the experiments. The description of the datasets and experimental setup is insufficient; it is notably unclear how gene lists from different datasets were integrated; especially without distinguishing up and down -regulated genes. Figures are difficult to read, and miss details in captions. The results section is often overly descriptive and lacks interpretative flow. Furthermore, the functional enrichment analyses do not seem to follow standard bioinformatic practices. The manuscript also includes numerous grammatical errors, unclear sentences, and poorly describe methods; making a thorough peer review challenging. Overall, while the topic and dataset choice are highly relevant, the manuscript requires substantial revision in writing, Figure presentation, and analytical rigor before I would recommend it for publication in International Journal of Molecular Sciences. Please see below for my major and minor points.

Major points

Comment of reviewer 1, Line128-138: This is an important part of the introduction as it should clearly state the objective of the study. Please clarify the overall goal: are the authors aiming to identify genes that are more highly expressed in cervical cancer compared to healthy controls, or to determine whether these genes are regulated by specific miRNAs? I would also kindly suggest including Figure 8 earlier in the manuscript as it nicely summarizes the workflow and would help readers understand the overall study aim.

Response: Thank you for the observation, the goal was added in this section as suggested by the reviewer.

Comment of reviewer 2, Figure1: The Venn diagram is difficult to interpret as it combines DEGs without distinguishing up and down -regulated genes. Some datasets may overlap due to genes significantly deregulated in the opposite direction. I would recommend separating up and down -regulated genes into two Venn diagrams. Furthermore, using a more classic round/oval Venn layout would also improve readability.

Response: Due to the selection of different microarrays and the intrinsic variability (number of genes, sample processing, etc.), it was decided to perform the analysis individually for each microarray. Once the data was obtained, the information was integrated (genes DEGs) using a Venn diagram with the aim of identifying those differentially expressed genes common to all microarrays, which allows for a global overview of genes that are expressed even when the sample conditions may be different, ssubsequently, the readings of each DEG (up and down) were extracted and compared using a color map for better contrast.

Comment of reviewer 3, Line151-156: I do not fully agree with the authors statement that higher number of samples provides “more accurate results” or leads to a higher number of DEGs. First, Table 1 does not support this claim, as GSE63514 includes only 52 samples (24 controls and 28 cancer), not 128, yet it reports many DEGs. Second, in cancer studies, sample quality, the choice of controls, and overall tissue heterogeneity (between cancer types, and healthy controls) have a lot of impact on DEG detection.

Response: Comparing GSE7691 versus GSE7803 we have 25 against 31 samples and a great difference in DEGs (632 vs 216), respectively.  The same applies for GSE63514 vs GSE39001 ((52 vs 55) (893 DEGs vs 249 DEGs)) and GSE9750 vs GSE52903 ((57 vs 72) (623 DEGs vs 342 DEGs)).

GSE7691

8 control and 17 cancer

632 DEGs

GSE7803

10 controls and 21 cancer

216 DEGs

GSE63514

24 controls and 28 cancer

893 DEGs

GSE39001

12 controls and 43 cancer

249 DEGs

GSE9750

24control and 33 cancer

623 DEGs

GSE52903

17 controls and 55 cancer

342 DEGs

Comment of reviewer 4, Figure2: Please increase font size and overall Figure quality. It is also not clear how the enrichment analysis was performed; notably, what input gene list was used?,

Response: All the genes that were up and down expressed in a least 4 microarrays, the exact genes can be founded in Table S1. 

Comment of reviewer 5, which significance threshold were applied? These details should be mention in Result section and/or Figure caption.

Response: The fold change was |1.5| and pa< 0.05

Comment of reviewer 6, Also, please explain the meaning of the asterisks shown next to some terms. Finally, what the ‘Transcription Factors’ corresponds to here?

Response: Thank you for the observation, asterisks were added in the processes were KEGG and Wikipathways share similar results. TF is product from the enrichment analysis.

Comment of reviewer 7, Method section mentioned that the enrichment analyses perform using enrichR, but which database has been used for ‘TF’?

Response: enrichR was used for TF enrichment analyses. Only the pathways important and the processes ontological significant p adjusted <0.05 were considered for the analysis.

Comment of reviewer 8, Another important point is that combining up and down -regulated genes in the same analysis may not be appropriate: please support this in the text, or consider separating up and down regulated genes in response to CC.

Response: Microarrays have intrinsic variability (number of genes, sample processing, etc.), therefore, we decided to perform the analysis individually for each microarray. Once the data was obtained, the information was integrated (genes DEGs) identifying those differentially expressed genes common to all microarrays, which allows for a global overview of genes that are expressed even when the sample conditions may be different, subsequently, the readings of each DEG (up and down) were extracted and compared.

Comment of reviewer 9, Line182-248: The result section reads more like a Discussion; there is excessive detail and interpretation, rather than concise reports of findings. The detailed functional descriptions of each TF may not be necessary, nor included here in the Result section. Please make this section more succinct, focusing on presenting key results only, and move biological interpretation to the Discussion.

Response: Thank you for the observation, we agree and the text was deleted and only the results was described.

Comment of reviewer 10, Furthermore, adding a short concluding statement summarizing the main takeway message would help for clarity and flow.

Response: Thank you for the observation, we agree and conclusion was added.

Comment of reviewer 11, The exact same comments apply for Line268-316.

Response: Thank you for the observation, we agree and the text was deleted and only the results was described.

Comment of reviewer 12, Figure3: Please increase font size and overall Figure quality. What does the color intensity correspond to? What type of values does this represent? Please add detail in Figure caption.

Response: Thank you for the observation, Red means upregulated and blue downregulated and was add in figure caption.

Comment of reviewer 13, Line258: How does the “prominently altered genes” have been selected?

Response: Thank you for the observation, Each of the genes that have 1.5 fold change and p adjusted <0.05 in each microarray and presented similar expression in a least 4 microarrays was used for the analysis. 

Comment of reviewer 14, Figure4: Please increase font size and overall Figure quality.

Response: Thank you for the observation, Figure quality was improved.

Comment of reviewer 15, Figure5: Please increase font size and overall Figure quality. Furthermore, it is unclear how the functional enrichment analysis was performed. Does the ten genes shown on the x-axis were used as input? What does each dot represent?; Does it indicate that the gene is included in a given ontology term? The figure and caption should clearly explain these points. The same clarification is also needed for Figure 7.

Response: Thank you for the observation, the font size and figure quality was improved. The function enrichment analysis was carried out in enrichR and the genes used were the up and downregulated of a least 4 microarrays sharing similar results (1.5 fold change and p adjusted <0.05). To determine the proteins of high biological value within the PPI networks groups (hub proteins), the hub option from the EnrichR platform (https://maayanlab.cloud/Enrichr/) was used. The Expression2Kinases program (Clarke et al., 2018) was used to identify regulatory proteins (mainly transcription factors (TFs) and kinases) involved in important signaling pathways that potentially regulate a PPI network based on the gene list submitted (Chen et al., 2012).

Comment of reviewer 16, Figure6: Please increase font size and overall Figure quality.

Response: Thank you for the observation, the font size and figure quality was improved.

Comment of reviewer 17, Line418-420: The three categories used by the authors to classify miRNAs are not clearly defined; please describe in more detail, notably, what is meant by “define profile”?

 Response: miRNAs detected experimentally by RT-PCR, microarrays, sequencing, norther-blot, etc.

Comment of reviewer 18, The Discussion section cannot be properly evaluated until the major issues outlined above are addressed.

Minor points

Comment of reviewer 19, Line145: Which data sheet? I was not able to access it.

Response: Thank you for the observation, the data sheet will share it.

Comment of reviewer 20, Table1: Please provide more detail on the sample if possible (ie. age of individuals, cancer stage, treatment, survival rate; any more information upon availabilities would be relevant to include here).

Response: Thank you for the observation, these data were added to the table as reviewer suggested.

Comment of reviewer 21, Line326-328: Are the Hub genes DE, or not? It is not clear what “not the first DEGs” means here.

Response: Yes, HUB genes resulted from a second analysis of ppi interaction and analysis were presented in figure 4.

Typos

Comment of reviewer 22, Line48: “associated to” seems incorrect; “in response to”?

Response: Thank you for the observation, the correct term is “associated to”, because no treatment is used.

Comment of reviewer 23, Line67-69: This sentence is not clear notably “that change transitory normal to immortal cell functions” please rephrase for clarity.

Response: transitory was deleted.

Comment of reviewer 24, Line78-80: What “is presented”? Please rephrase for clarity.

Response:  Thank you for the observation, “is presented” was deleted.

Comment of reviewer 25, Line129-133: Repetitive, please rephrase.

Response: Thank you for your observation, the sentences “Recent studies have utilized microarray data to identify DEGs associated with cervical cancer, uncovering potential biomarkers for early detection and therapeutic targets” was deleted to override repetitive paragraph.

 Comment of reviewer 26, Line178: What “can be advised”? Please rephrase for clarity.

Response: Thank you for your observation, “can be advised” was deleted.

 Comment of reviewer 27, Line183-185: This sentence is not clear, please rephrase.

Response: Notably, the cellular component analysis was associated with cell death, protein binding, DNA replication, JUN factor transcription, and several miRNAs (miR-192-5p, miR-193b, and miR-215-5p) as the most prominent changes among the 6 microarrays.

Comment of reviewer 28, Line203-205: “International Federation of Gynecology and Obstetrics”?

Response: The stage of International Federation of Gynecology and Obstetrics stages have 4 stages:

Comment of reviewer 29, Line249: Title missing for 2.1.2

Response: Thank you for your observation, the title “Microarray analysis integration and common genes determination” was added.

Comment of reviewer 30, Line264: “they did not appear as DEGs” ; miss a word

Response: Thank you for the observation, the missing word is “genes” and were added to the manuscript. 

Comment of reviewer 31, Line323: Figure4, and not Figure3.

Response: Thank you for your observation, the suggested change was added.

Comment of reviewer 32, Line382-406: Part “Functional enriched process for HUB proteins” is duplicated.

Response: Thank you for your observation, “Functional enriched process for HUB proteins” was deleted.

Comment of reviewer 33, Line469: “whit”

Response: Thank you for the observation, whit was corrected.

Comment of reviewer 34, Line479: Please rephrase.

Response: Several paragraphs were deleted, this could be revised in the manuscript.

Reviewer 2 Report

Comments and Suggestions for Authors

The manuscript “Differentially expressed genes associated to the development of cervical cancer” presents an integrative bioinformatics analysis combining six GEO datasets to identify differentially expressed genes (DEGs) and miRNA–mRNA regulatory networks in cervical cancer. The topic is relevant and timely as it advances molecular knowledge with potential clinical significance. However, there are several methodological issues and questions regarding the patient databases used that need to be clarified to lend greater credibility to the study.

Major comments                                                                                                

  1. Novelty

Although the authors use several datasets in their analysis, they do not justify what the new contribution of this study is when compared to previous works with a similar approach, which even analyzes some GSEs used in this study. For example: (GSE 7803)     ; (GSE 9750 and 63514) https://doi.org/10.3389/fgene.2021.775006; (GSE 63514) https://doi.org/10.1186/s12885-024-12658-z .

  1. GSE selection and clinical-pathological characteristics patients

While figure 8 shows a workflow indicating dataset selection, the criteria used to select these specific GSE datasets are not explained. It is unclear whether inclusion/exclusion criteria were based on platform compatibility, population heterogeneity, sample size, FIGO stage, clinical data availability, histological homogeneity, or other relevant clinical variables.

Although in the introduction (line 129-138) the authors briefly comment on the heterogeneity of the samples included in the six datasets analyzed, the manuscript does not provide any clinical description of the patient cohorts included. There is no table summarizing essential characteristics such as age, FIGO stage, tumor histology, HPV genotype, or treatment outcome. This information is fundamental to properly contextualizing gene expression differences in cervical cancer.

I strongly recommend: Including a table of cohort characteristics or described in materials and methods. Describing clear selection criteria for dataset inclusion in methods and discussing the potential impact of clinical heterogeneity on the resulting DEGs. If the authors clearly justify the dataset selection criteria and define the novelty and relevance of their integrative analysis, the scientific basis of this study will be significantly strengthened.

  1. Data analysis

The authors argue that analyzing each microarray independently using GEO2R helps avoid batch effects. This is only partially correct: while separate processing prevents artificial batch introduction from merging raw data, it does not eliminate inherent heterogeneity across studies, including differences in preprocessing, patient populations, etc. Is this type of analysis sufficient, or is it better to perform a meta-analysis and apply another type of data processing?

  1. Data validation

Studies of this type are commonly validated with TCGA data or experimental validation of some of the findings. These results are even more significant if a hub protein or miRNA is associated with a clinical characteristic, survival, or response to treatment. Can the authors supplement their analysis with any of these options to support their results? To provide evidence that the identified molecules have potential as biomarkers.

  1. Discussion

In general, the results contain a lot of information and comparisons that should be included in the discussion. The authors need to restructure both sections so as not to repeat information and to enrich the discussion. It is also necessary to mention the limitations and scope of the research.

Line 461-462: “In this study the bioinformatics analyses performed on six cervical cancer miRNAs expression´s microarray studies…” Is the data only from miRNAs? Why start the discussion by talking about miRNAs? Are they the most important findings in this study?

  1. Figures

All figures must be of higher quality for publication.

Some figures, such as figure 6, are practically incomprehensible due to their low quality.

Minor comments

Line 430: The correct figure is 6A and 6B.

Line 583: CC is used as an abbreviation for cellular components, and CC is already used for cervical cancer.

Add a brief conclusion.

Author Response

Dear reviewer, I tried to respond to all suggestions and observations.

The manuscript “Differentially expressed genes associated to the development of cervical cancer” presents an integrative bioinformatics analysis combining six GEO datasets to identify differentially expressed genes (DEGs) and miRNA–mRNA regulatory networks in cervical cancer. The topic is relevant and timely as it advances molecular knowledge with potential clinical significance. However, there are several methodological issues and questions regarding the patient databases used that need to be clarified to lend greater credibility to the study.

Major comments                                                                                               

Novelty

Comment of reviewer 1, Although the authors use several datasets in their analysis, they do not justify what the new contribution of this study is when compared to previous works with a similar approach, which even analyzes some GSEs used in this study. For example: (GSE 7803)     ; (GSE 9750 and 63514) https://doi.org/10.3389/fgene.2021.775006; (GSE 63514) https://doi.org/10.1186/s12885-024-12658-z .

Response: Thank you for the observation, the generation of all the work with miRNAs-mRNAs. 

Comment of reviewer 2, GSE selection and clinical-pathological characteristics patients

While figure 8 shows a workflow indicating dataset selection, the criteria used to select these specific GSE datasets are not explained. It is unclear whether inclusion/exclusion criteria were based on platform compatibility, population heterogeneity, sample size, FIGO stage, clinical data availability, histological homogeneity, or other relevant clinical variables.

Response: Due to the selection of different microarrays and the intrinsic variability (number of genes, sample processing, etc.), it was decided to perform the analysis individually for each microarray. Once the data was obtained, the information was integrated (genes DEGs) using a Venn diagram with the aim of identifying those differentially expressed genes common to all microarrays, which allows for a global overview of genes that are expressed even when the sample conditions may be different, ssubsequently, the readings of each DEG (up and down) were extracted and compared using a color map for better contrast.

Comment of reviewer 3, Although in the introduction (line 129-138) the authors briefly comment on the heterogeneity of the samples included in the six datasets analyzed, the manuscript does not provide any clinical description of the patient cohorts included. There is no table summarizing essential characteristics such as age, FIGO stage, tumor histology, HPV genotype, or treatment outcome. This information is fundamental to properly contextualizing gene expression differences in cervical cancer.

Response: Thank you for your observation, Table 1 was enriched with this information.

Comment of reviewer 4, I strongly recommend: Including a table of cohort characteristics or described in materials and methods. Describing clear selection criteria for dataset inclusion in methods and discussing the potential impact of clinical heterogeneity on the resulting DEGs. If the authors clearly justify the dataset selection criteria and define the novelty and relevance of their integrative analysis, the scientific basis of this study will be significantly strengthened.

Response: The selection of the six GEO datasets (GSE39001, GSE9750, GSE7803, GSE6791, GSE63514, and GSE52903) was selected based on predefined inclusion criteria emphasizing data completeness, biological relevance, and methodological comparability. Although other GEO datasets on cervical cancer are available, many did not meet these criteria because they lacked matched normal controls, contained very small sample sizes, or used platforms with limited probe annotation or inconsistent normalization. The selected series collectively provide one of the most balanced combinations of sample depth, clinical annotation, and technological quality available in GEO for cervical cancer. Moreover, they encompass both Affymetrix and Agilent platforms two well standardized microarray systems allowing coverage of a wide molecular spectrum while maintaining data reliability. Recognizing that multi-platform integration introduces potential batch effects from differences in probe design and chemistry, we intentionally avoided direct data merging and instead conducted independent differential expression analyses using GEO2R (FDR < 0.05, |logFC| > 1.5). Only genes reproducibly dysregulated in at least four of the six datasets were retained. This intersection based approach substantially reduces inter platform noise and emphasizes consistently deregulated genes across independent cohorts, thereby maximizing analytical robustness without introducing additional normalization bias. From this high confidence DEG core, hub genes were identified using STRING-based PPI construction and cytoHubba’s Maximal Clique Centrality (MCC) algorithm, ensuring that both expression consistency and network relevance guided the prioritization. The resulting hubs KIF4A, NUSAP1, BUB1B, CEP55, DLGAP5, NCAPG, CDK1, MELK, KIF11, and KIF20A were consistently upregulated across datasets and mapped to mitotic spindle assembly, chromosomal segregation, and cell cycle pathways, all fundamental to tumor progression. Their recurrence across independent cohorts supports their role as biologically conserved regulators rather than dataset specific artifacts. Complementarily, miRNAs interacting with multiple hub genes were retrieved from miRNet and RNA-Disease databases, restricted to experimentally validated interactions and to miRNAs previously implicated in cervical cancer. The observed inverse expression trends such as downregulated tumor suppressive miRNAs (miR-107, miR-124-3p, let-7b-5p) alongside upregulated mitotic hubs support a plausible post-transcriptional regulatory model. Collectively, the dataset choice, the built-in strategy to limit batch variability, and the integration of validated mRNA–miRNA interactions form a rigorous and reproducible framework that meets current standards for transparency, robustness, and biological interpretability in integrative cancer genomics research.

Comment of reviewer 5, The authors argue that analyzing each microarray independently using GEO2R helps avoid batch effects. This is only partially correct: while separate processing prevents artificial batch introduction from merging raw data, it does not eliminate inherent heterogeneity across studies, including differences in preprocessing, patient populations, etc. Is this type of analysis sufficient, or is it better to perform a meta-analysis and apply another type of data processing?

Response: Although a meta-analysis is recommended for comparisons between microarrays, variables such as the batch effect, sample number, gene microarray probes, platform type, and RNA ca-pacity detection complicate accurate statistical analysis according to several studies as (Ghosh et al., 2003; Ramasamy et al., 2008). For this reason, we chose to analyze each platform independently. By contrasting the overall context of different microarrays, we identified DEGs in common for each microarray with similar expression patterns, which are shown in the heatmap. Only genes present in at least 4 of the 6 microarrays were selected for further studies.

Comment of reviewer 6, Data validation

Studies of this type are commonly validated with TCGA data or experimental validation of some of the findings. These results are even more significant if a hub protein or miRNA is associated with a clinical characteristic, survival, or response to treatment. Can the authors supplement their analysis with any of these options to support their results? To provide evidence that the identified molecules have potential as biomarkers.

Response: Thank you for the observation. Survival analysis was done in UALCAN data base.

Comment of reviewer 7, Discussion

In general, the results contain a lot of information and comparisons that should be included in the discussion. The authors need to restructure both sections so as not to repeat information and to enrich the discussion. It is also necessary to mention the limitations and scope of the research.

Response: Thank for the observation, the limitations will be added in the discussion has suggested by the reviewer.

Comment of reviewer 8, Line 461-462: “In this study the bioinformatics analyses performed on six cervical cancer miRNAs expression´s microarray studies…” Is the data only from miRNAs? Why start the discussion by talking about miRNAs? Are they the most important findings in this study?

Response: Thank for the observation, the discussion while started with mRNAs expression of all the finding of our results. 

Comment of reviewer 9, Figures

All figures must be of higher quality for publication.

Some figures, such as figure 6, are practically incomprehensible due to their low quality.

Responses: Thank you for the observation, all the figures were improved.

Minor comments

Comment of reviewer 10, Line 430: The correct figure is 6A and 6B.

Response: Thank you for the observation, Figure was corrected.

Comment of reviewer 11, Line 583: CC is used as an abbreviation for cellular components, and CC is already used for cervical cancer.

Response: Thank you for the observation, CC was substituted for cellular components.

Comment of reviewer  12, Add a brief conclusion.

Response: Thank you for the observation, a conclusion was added.

Round 2

Reviewer 1 Report

Comments and Suggestions for Authors

I thank the authors for providing a revised version of their manuscript. While some of my previous points were addressed, this new version of the manuscript still suffers from significant issues, notably with the experimental design, data integration, readability of Figures, and scientific flow. I appreciate that the authors now clearly stated the goal of the study; however, the goal does not align with the analyses they perform. Notably, the authors state that their objective is to “find genes that are more highly expressed in cervical cancer compared to healthy control.” However, when integrating the DGEs across the different microarray, they do not separate up and down -regulated genes. This can easily generate false positives and is not aligned with their research objective. Indeed, with the aim to identify genes consistently upregulated in cervical cancer, then integrating DEGs without separating directionality is not appropriate. I strongly recommend that the authors re-think the experimental design and candidate selection, focusing exclusively on up-regulated genes across datasets. If they choose to include both up and down -regulated genes, then they must clearly justify why this is biologically meaningful and how contradictory regulation (up in one dataset, down in another) can be considered a reliable biomarker? The manuscript also remains unclear in multiple sections. For example, the authors mention in their response that they quantified miRNAs by RT-PCR, but these results are not shown and are not described in the Methods. If such experiments were performed, they should be properly included. Unfortunately, while the topic and dataset choice are highly relevant I remain unconvinced by the overall design, analyses, and clarity of the manuscript. For these reasons, I cannot recommend this paper for publication, nor can perform a complete thorough peer-review.

Major points

Figure2: The venn diagram is still not clear, the authors did not address my previous comments. As stated earlier, I strongly recommend separating up and down -regulated genes, notably as the study objective clearly state that the authors look for genes “more highly expressed in cervical cancer compared to healthy control”. Also please consider changing the visualization using a classic round/oval Venn layout to improve readability. Same comment applies for Figure3 for readability.

Line193-198: I still disagree with the authors statement that the number of DEGs is link with the number of samples. The response provided is confusing, as it compares data selectively one by one: without looking at the full picture: Please see below samples ranked from top to low number of DEGs:

  • GSE63514= 52 total samples= 893 DEGs
  • GSE7691= 25 total samples= 632 DEGs
  • GSE9750= 57 total samples= 623 DEGs
  • GSE52903= 72 total samples= 342 DEGs
  • GSE39001= 55 total samples= 249 DEGs
  • GSE7803= 22 total samples= 216 DEGs

It is clear that dataset with the highest number of samples do not rank among those with the highest DEGs…. Interestingly, even GSE7691 with only 25 samples rank among the top. Please revise your statement accordingly.

The Result and Discussion section cannot be properly evaluated until the major issues outlined above are addressed.

Author Response

Dear Reviewer 1 (round 2)

Our work group appreciates the time, expertise, and constructive feedback provided during the review process. Your comments have been significant and have contributed with scientific rigor, clarity, and overall quality to our manuscript. We carefully considered each observation and provided a response indicating the changes implemented in the revised version. We hope these revisions enhance the value and transparency of our study.

Major points

Figure2: The venn diagram is still not clear, the authors did not address my previous comments. As stated earlier, I strongly recommend separating up and down -regulated genes, notably as the study objective clearly state that the authors look for genes “more highly expressed in cervical cancer compared to healthy control”. Also please consider changing the visualization using a classic round/oval Venn layout to improve readability. Same comment applies for Figure3 for readability.

Answer 1: Thank you for your observation. Based on your suggestions, we updated Figure 2, displaying two diagrams showing up- and down-regulated genes. We hope this new figure illustrates the distribution of common DEGs across microarray comparisons.

Line193-198: I still disagree with the authors statement that the number of DEGs is link with the number of samples. The response provided is confusing, as it compares data selectively one by one: without looking at the full picture: Please see below samples ranked from top to low number of DEGs:

  • GSE63514= 52 total samples= 893 DEGs
  • GSE7691= 25 total samples= 632 DEGs
  • GSE9750= 57 total samples= 623 DEGs
  • GSE52903= 72 total samples= 342 DEGs
  • GSE39001= 55 total samples= 249 DEGs
  • GSE7803= 22 total samples= 216 DEGs

It is clear that dataset with the highest number of samples do not rank among those with the highest DEGs…. Interestingly, even GSE7691 with only 25 samples rank among the top. Please revise your statement accordingly.

Answer 2: Thank you for your observation. I agree with you, samples with highest number of samples have less DEGs that’s what a I observed and thats what I tried to write in the manuscript, however, we analyzed only 6 microarrays, therefore, we think we have to analyze more assays to assert this observation.

Based on this consideration the line “It can be implied that the number of samples evaluated in the studies may influence the number of DEGs detected” was removed from the manuscript according to observation. 

The Result and Discussion section cannot be properly evaluated until the major issues outlined above are addressed.

Answer 3: Thank you, and we hope to receive your observations and comments on the rest of the manuscript soon.

Reviewer 2 Report

Comments and Suggestions for Authors

Thanks to the authors for correcting most of the comments. Based on the improved version of the manuscript, I am sending the following comments and suggestions.

Major comments

1. Rewrite or match the objective in lines 136-138, as it differs from what is stated in the summary (lines 39-41).

2. Materials and methods. Thank you for clarifying how the datasets were selected ("The selection of the six GEO datasets (GSE39001, GSE9750, GSE7803, GSE6791, GSE63514, and GSE52903) was selected based on predefined inclusion criteria emphasizing data completeness, biological relevance, and methodological comparability. Although other GEO datasets on cervical cancer are available, many did not meet these criteria because they lacked matched normal controls, contained very small sample sizes, or used platforms with limited probe annotation or inconsistent normalization. The selected series collectively provide one of the most balanced combinations of sample depth, clinical annotation, and technological quality available in GEO for cervical cancer. Moreover, they encompass both Affymetrix and Agilent platforms two well standardized microarray systems allowing coverage of a wide molecular spectrum while maintaining data reliability..."). This information should be properly integrated into the materials and methods, as although Figure 1 mentions that a selection was performed, it does not explain how it was done.

4. Survival analysis: Although survival curves are presented with the selected genes, none are statistically significant. Therefore, this does not agree with what is described in lines 402-405. Please confirm the importance of these genes with other analyses. If there is no correlation with overall survival, perhaps you could confirm whether some of your hub genes can be validated with TCGA data.

5. Discussion. The discussion section needs improvement. First, the authors begin by discussing the processes with which miRNAs are associated. This study is an analysis of various public data, so they should begin the discussion by emphasizing the importance and advantages of this type of reanalysis. Although the results are interesting and the discussion makes biological sense, there is no order to the discussion of the different sections presented in the results. I suggest using the results as a guide to present a more structured discussion.

7. The conclusion needs to be summarized to present only the key findings. It contains statements that correspond to the discussion section.

Minor comments

The paragraph on lines 436-449 contains information for discussion.

Line 492-493: “Furthermore, miR-10b-5p, miR-23b-3p, miR-449a, and miR-130a-3p were found modulating 8 and 7 of 10 HUBs genes, respectively.” There are four microRNAs listed, which regulate 8 and 7 hub genes?

A section on limitations has not yet been added to the discussion.

Author Response

Dear Reviewer 2.

Our work group appreciates the time, expertise, and constructive feedback provided during the review process. Your comments have been significant and have contributed with scientific rigor, clarity, and overall quality to our manuscript. We carefully considered each observation and provided a response indicating the changes implemented in the revised version. We hope these revisions enhance the value and transparency of our study.

Major comments

  1. Rewrite or match the objective in lines 136-138, as it differs from what is stated in the summary (lines 39-41).

Answer 1: Thank you for your observation, we rewrote the manuscript to specify that a DEG determination in the CaCu microarray and its subsequent ontology analysis was realized, and this includes a bioinformatics determination of possible associated miRNAs.

  1. Materials and methods. Thank you for clarifying how the datasets were selected ("The selection of the six GEO datasets (GSE39001, GSE9750, GSE7803, GSE6791, GSE63514, and GSE52903) was selected based on predefined inclusion criteria emphasizing data completeness, biological relevance, and methodological comparability.

Although other GEO datasets on cervical cancer are available, many did not meet these criteria because they lacked matched normal controls, contained very small sample sizes, or used platforms with limited probe annotation or inconsistent normalization. The selected series collectively provide one of the most balanced combinations of sample depth, clinical annotation, and technological quality available in GEO for cervical cancer. Moreover, they encompass both Affymetrix and Agilent platforms two well standardized microarray systems allowing coverage of a wide molecular spectrum while maintaining data reliability...").

This information should be properly integrated into the materials and methods, as although Figure 1 mentions that a selection was performed, it does not explain how it was done.

Answer 2: Thank you for your observation, but this paragraph was a suggestion from reviewer 1, and we believed that this information is useful in the results as well as in materials section as you suggested.

  1. Survival analysis: Although survival curves are presented with the selected genes, none are statistically significant. Therefore, this does not agree with what is described in lines 402-405. Please confirm the importance of these genes with other analyses. If there is no correlation with overall survival, perhaps you could confirm whether some of your hub genes can be validated with TCGA data. Figura 5 volarlo y figura 6

Answer 3: Thanks for your observation. We performed a new revision directly on the TCGA platform and found no significant correlation in survival analysis, so we decided to omit this analysis from the manuscript. Unfortunately, the reports of cervical cancer are still limited, and our DEGs are only significantly linked with other cancer types in a survival context.

  1. Discussion. The discussion section needs improvement. First, the authors begin by discussing the processes with which miRNAs are associated. This study is an analysis of various public data, so they should begin the discussion by emphasizing the importance and advantages of this type of reanalysis. Although the results are interesting and the discussion makes biological sense, there is no order to the discussion of the different sections presented in the results. I suggest using the results as a guide to present a more structured discussion.

Answer 4: Thanks for your observation. The discussion section was improved, and the miRNAs discussion was rewritten and organized according to the results.

  1. The conclusion needs to be summarized to present only the key findings. It contains statements that correspond to the discussion section.

Answer 5: We have rewritten the conclusion in a summary form, including only the results we consider significant.

Minor comments

The paragraph on lines 436-449 contains information for discussion.

Answer 6: Thanks for your observation. The information in paragraphs 436-449 was moved to the discussion section, as you suggested, because it concerned gene functions. 

Line 492-493: “Furthermore, miR-10b-5p, miR-23b-3p, miR-449a, and miR-130a-3p were found modulating 4, 4, 6, and 7 of 10 HUBs genes, respectively.” There are four microRNAs listed, which regulate 8 and 7 hub genes?

Answer 7: Thanks for your observation. We corrected this section, and the correct paragraph is “Furthermore, miR-10b-5p, miR-23b-3p, miR-449a, and miR-130a-3p were found modulating 4, 4, 6, and 7 of 10 HUBs genes, respectively.”

A section on limitations has not yet been added to the discussion.

Answer 8: Thanks for your observation. We aggregated this section into the manuscript according on your observations.

Round 3

Reviewer 2 Report

Comments and Suggestions for Authors

Thank you for considering the comments made. The work currently presented has potential.
I inadvertently omitted one additional point (Point 3) in my last review, which I now include below for your consideration:

3. Clinical data

Table 1 presents clinical data that needs to be reviewed. For example:

GSE39001: Clinical data are disponible in supporting information of Espinosa AM, Alfaro A, Roman-Basaure E, Guardado-Estrada M et al. Mitosis is a source of potential markers for screening and survival and therapeutic targets in cervical cancer. PLoS One 2013;8(2):e55975. PMID: 23405241

GSE9750: The stage reported in Table 1 is wrong. After checking the first five samples in this dataset (GSM247650 Cervical cancer_CC126, GSM247651 Cervical cancer_CC128, GSM247652 Cervical cancer_CC140, GSM247653 Cervical cancer_CC163, GSM247654 Cervical cancer_CC205), the samples are stage IIIB. Please carefully review the information and report the correct stage.

The same applies to the other datasets. I strongly recommend carefully reviewing the available clinical data. I suggest presenting all clinical data in a supplementary table to clearly visualize the percentages of stages, HPV positivity, etc.

Finally, I continue to believe and suggest that it is relevant to mention in the discussion the scope of your analysis due to the sample size analyzed, given the six cohorts used. What advantages does it have over previous studies, and how does this impact the discovery of new molecules associated with cervical carcinogenesis?

Author Response

Dear reviewer, we truly appreciate the time you dedicate to reviewing our work.

Comments 1. Thank you for considering the comments made. The work currently presented has potential.
I inadvertently omitted one additional point (Point 3) in my last review, which I now include below for your consideration:

  1. Clinical data

Table 1 presents clinical data that needs to be reviewed. For example:

Response 1:

Dear reviewer, thank you for your last observation, and we appreciate the time and interest you have shown in our work.

We checked and compared the information reported in the supply material for each microarray. However, some of the supplied material was incomplete for some papers (GSE63514 and partially GSE52903).

So we decided to analyze each sample in the GEO database for each microarray and collect three common characteristics (VPH type, histology, and tumor stage) (according to your suggestion); however, data for GSE63514 were unavailable because the authors provided only a summary table in the supplementary material, and the GEO database doesn’t show complete information from this microarray.

However, based on our tracking information, we provide a table with general data collected from the control and CaCu samples used in our work across different microarrays, a second table containing only CaCu data, and the GSE63514 supply table, as provided by the authors. These data were analyzed to identify the three most common variables across studies (vph type, histology, and tumor stage), and we report the trends identified.

Comments 2: Finally, I continue to believe and suggest that it is relevant to mention in the discussion the scope of your analysis due to the sample size analyzed, given the six cohorts used. What advantages does it have over previous studies, and how does this impact the discovery of new molecules associated with cervical carcinogenesis?

Response 2:

To incorporate your valuable comments and connect the conclusion with the clinical information, we rewrote the limitations section of our work.  We included a paragraph on the different stages and VPH types in CaCu samples, which may need to be checked and contrasted in future similar studies.